# Time-Restricted Feeding Attenuates Metabolic Dysfunction-Associated Steatohepatitis and Hepatocellular Carcinoma in Obese Male Mice

**DOI:** 10.3390/cancers16081513

**Published:** 2024-04-16

**Authors:** Manasi Das, Deepak Kumar, Consuelo Sauceda, Alexis Oberg, Lesley G. Ellies, Liping Zeng, Lily J. Jih, Isabel G. Newton, Nicholas J. G. Webster

**Affiliations:** 1VA San Diego Healthcare System, San Diego, CA 92161, USA; madas@health.ucsd.edu (M.D.);; 2Department of Medicine, Division of Endocrinology and Metabolism, University of California San Diego, La Jolla, CA 92093, USA; 3Department of Pathology, University of California San Diego, La Jolla, CA 92093, USA; 4Moores Cancer Center, University of California San Diego, La Jolla, CA 92093, USA; 5Department of Radiology, University of California San Diego, La Jolla, CA 92093, USA

**Keywords:** non-alcoholic fatty liver disease, metabolic dysfunction-associated steatohepatitis, hepatocellular cancer, dietary intervention, time-restricted feeding, mouse model

## Abstract

**Simple Summary:**

Obesity and metabolic syndrome are major drivers of metabolic liver disease and liver cancer. While caloric restriction can abrogate many of the deleterious effects of obesity, weight loss through dieting is not sustainable in the general population. We have investigated whether an alternative approach limiting the window of eating, rather than the caloric content, could have similar beneficial effects. We found that restricting food intake to an eight-hour window during the active phase improved liver metabolism and reduced liver cancer in preclinical models of obesity-driven liver cancer.

**Abstract:**

Metabolic dysfunction-associated steatotic liver disease (MASLD) has surpassed the hepatitis B virus and hepatitis C virus as the leading cause of chronic liver disease in most parts of the Western world. MASLD (formerly known as NAFLD) encompasses both simple steatosis and more aggressive metabolic dysfunction-associated steatohepatitis (MASH), which is accompanied by inflammation, fibrosis, and cirrhosis, and ultimately can lead to hepatocellular carcinoma (HCC). There are currently very few approved therapies for MASH. Weight loss strategies such as caloric restriction can ameliorate the harmful metabolic effect of MASH and inhibit HCC; however, it is difficult to implement and maintain in daily life, especially in individuals diagnosed with HCC. In this study, we tested a time-restricted feeding (TRF) nutritional intervention in mouse models of MASH and HCC. We show that TRF abrogated metabolic dysregulation induced by a Western diet without any calorie restriction or weight loss. TRF improved insulin sensitivity and reduced hyperinsulinemia, liver steatosis, inflammation, and fibrosis. Importantly, TRF inhibited liver tumors in two mouse models of obesity-driven HCC. Our data suggest that TRF is likely to be effective in abrogating MASH and HCC and warrant further studies of time-restricted eating in humans with MASH who are at higher risk of developing HCC.

## 1. Introduction

A sedentary lifestyle coupled with a high-fat, calorie-rich Western diet (WD) induces obesity and can lead to metabolic dysfunction-associated steatotic liver disease (MASLD) [1]. Due to the increase in obesity in the USA, the number of people with MASLD is rapidly increasing and is expected to reach 1/3rd of the USA’s population by 2030 [2,3]. MASLD ranges from simple fatty liver, characterized by hepatic steatosis, to more advanced metabolic dysfunction-associated steatohepatitis (MASH) with inflammation, hepatocyte ballooning, and different degrees of fibrosis [4]. Approximately 10–20% of people with MASLD have MASH, and a subset of those will progress to liver cirrhosis and its associated complications, such as hepatocellular carcinoma (HCC). Although the early stages of MASLD are thought to be reversible, more advanced MASH and cirrhosis cause irreversible damage.

Hepatocellular carcinoma (HCC) is the sixth most prevalent cancer globally and the second major cause of cancer-related deaths [5]. HCC accounts for 70–85 percent of primary liver cancer and can be of viral and non-viral origin. The last few decades have witnessed the emergence of non-viral HCC as the major cause of liver cancer in the US. Indeed, the population-attributable fraction for obesity-related HCC in the US is 37% [6], much higher than HCV (22%), HBV (6%), and alcohol-related HCC (24%). Although the risk of hepatocellular carcinoma (HCC) increases significantly for MASLD or MASH patients who have cirrhosis [7,8], approximately 20% of patients with HCC do not have cirrhosis, and there is increasing evidence that HCC can develop in the setting of MASLD [9]. Steatosis alone can promote carcinogenesis, as obesity and metabolic syndrome are independent risk factors for the development of HCC, with a 1.5–4 times increased risk [7]. This risk is likely conferred by two factors: the increased risk of subsequent development of MASLD/MASH and the carcinogenic potential of obesity alone. As MASLD, MASH, and HCC are interlinked and strongly associated with obesity [10,11,12,13,14], strategies that mitigate obesity and the associated metabolic dysfunction may reduce the risk of MASH progression and HCC. 

Caloric restriction or fasting to reduce weight in obese individuals has beneficial effects on metabolism, MASLD, and cancer growth in both mice and humans [15,16,17]. Despite these encouraging outcomes, weight loss strategies have not been successful in the general population, as they trigger hunger and irritability and can involve calorie counting, dietary planning, or exercise, which can be difficult to incorporate into a busy lifestyle, especially for people being treated for HCC, all of which limits long-term adherence [18,19]. As an alternative to dieting, restricting when calories are ingested (time-restricted eating, TRE) rather than the quantity or quality of calories is an attractive option [20,21,22,23]. Time-restricted eating is a dietary regimen in which food intake is restricted to a particular number of hours per day, typically 6–12 h in alignment with circadian rhythms and without any calorie reduction or change in diet [20,24,25,26,27,28,29]. TRE is gaining popularity as a new intervention for improved metabolic health and weight control. Studies in mice have demonstrated the beneficial effect of TRE in protecting and reversing some of the cardiometabolic effects of obesity and have also shown reductions in inflammation and cancer [21,22,23,25]. The effect of TRE on metabolic liver disease is less well studied. In preclinical murine models of MASLD, TRE improves inflammation without weight loss [30], and a recent study demonstrated that TRE led to significantly lower body weight and MASLD activity score and improved both MASH and liver fibrosis [31]. Human studies are less clear, however, depending on the groups studied. Recent meta-analyses of intermittent fasting studies in MASLD indicated that this intervention reduces body weight, liver enzymes, steatosis, and liver stiffness, particularly in overweight or obese individuals [32,33,34,35,36,37]. Another cross-sectional study reported that an eating window of <8 h was associated with a lower risk of MASLD [38]. As an intervention, improvements in MASH have been noted during Ramadan in which people restrict their eating time to nighttime for 30 days [39,40]. The results demonstrate that daytime fasting significantly improves non-invasive markers of fatty liver disease (including Fibrosis-4 Index (FIB-4) score, MASLD Fibrosis Score, and BARD Score), reduces insulin resistance, induces weight loss, and improves inflammatory markers [41]. Although the ability of daytime fasting to abrogate MASH looks promising, nighttime fasting in alignment with the circadian clock may provide even greater metabolic benefits.

Given the promising anti-tumor benefits of TRE that we had shown previously in models of obesity-driven postmenopausal breast cancer in female mice, we wanted to investigate whether TRE would have similar beneficial effects in MASH-driven HCC. In this study, we aimed to demonstrate the effects of TRE on the pathogenic changes associated with MASH, including glucose and insulin sensitivity and hyperinsulinemia, and, more importantly, on liver tumor growth in two models of MASH-driven HCC. 

## 2. Materials and Methods

### 2.1. Animals, Diets, and Housing Conditions

Male C57Bl/6J mice were purchased from Jackson Laboratories. Mice were housed at 22 ± 2 °C on a 12 h light/12 h dark cycle (lights on at 6 a.m.) with free access to food and water prior to being placed in study groups. A graphic representation of the time-restricted fasting (TRF, equivalent of TRE in humans) strategy is shown in Figure 1A. For the metabolic study, sixteen male C57BL/6J mice were started on the WD at 8 weeks of age. A separate group of eight C57BL/6J mice was maintained on the NC diet as a lean control group. Mice were placed on a Western diet (WD; 40% milk fat, 29% sucrose, 0.2% cholesterol; D12079B, Research Diets) to cause MASLD and MASH [42]. Control mice were fed a normal chow diet (NC; rodent 5001, Research Diets). Male mice were used as females tend to be more resistant to MASLD. Body weight and food intake were measured weekly thereafter. When the mice on the WD reached an average weight of 45 g (~10 weeks of diet), the WD group was divided into ad libitum (AL) and time-restricted feeding (TRF) groups of 8 mice each continuing on the WD. Custom fully-automated feeder cages were used for the AL and TRF intervention study with the TRF group having access to food from 8 p.m. to 4 a.m. daily (Figure 1B), while the AL group had access to the diet at all times to control for cage effects [23]. All animal studies were reviewed and approved by the UCSD Institutional Animal Care and Use Committee.

### 2.2. Spontaneous Tumor Model

Mice with a hepatocyte-specific deletion of serine and arginine-rich splicing factor 3 (SRSF3-HKO) were used as a spontaneous HCC model. SRSF3 has been implicated in controlling hepatocyte differentiation, glucose and lipid metabolism, and many other signaling pathways [43]. We have also reported that genetic deletion of SRSF3 in hepatocytes causes fibrosis, steatohepatitis, and the development of metastatic HCC with aging, with mutational signatures similar to human HCC [44]. Furthermore, SRSF3 expression is decreased in humans with liver disease or HCC [45]. So, the SRSF3-HKO mouse model is very useful for understanding human liver disease in the context of obesity or MASH. Ten SRSF3-HKO mice at 8 weeks of age were placed on the WD and randomly divided into AL and TRF groups. Livers were assessed for tumors at 28 weeks of age by high-frequency ultrasound (Vevo-MD). At the end of this study, mice were euthanized, plasma was collected, and tumors, visceral fat, and livers were harvested and weighed. Tissues were formalin fixed or frozen for further analysis.

### 2.3. Subcutaneous Hepa1-6 HCC Model

Hepa1-6 mouse hepatocarcinoma cells (CRL-1830, ATCC, Manassas, VA, USA) were purchased and maintained in high glucose DMEM, with 10% FBS and 1× penicillin–streptomycin, in a humidified 5% CO_2_ incubator at 37 °C. Cells were passaged twice a week. For the tumor growth study, 16 male C57BL/6J mice were started on WD and 8 were maintained on NC. After 8 weeks, the mice on the WD were randomized to TRF or AL groups. After 4 weeks from the start of TRF, subcutaneous liver tumors were generated by injecting Hepa1-6 cells (12,000 cells) in a 20 μL matrigel/PBS solution (1:1 cells in PBS/matrigel) bilaterally into the left and right flank. Tumors were palpated weekly, and dimensions were measured with calipers once tumors were detected. At the end of this study, mice were euthanized, plasma was collected, and tumors, visceral fat, and livers were harvested and weighed. Tissues were fixed in 10% neutral-buffered formalin or frozen for further analysis as before.

### 2.4. Ultrasound Methods

A VevoMD ultra-high-frequency ultrasound system equipped with a 70 MHz probe (FUJIFILM VisualSonics, Toronto, ON, Canada) was used to image the liver. Animals were sedated in a supine position under isofluorane anesthesia delivered via a nose cone. The ventral fur was removed with chemical depilatory cream, and ultrasound gel was applied to the abdomen. Tumors appeared as a well-circumscribed, hypoechoic mass. Dimensions were measured in two axes in the transverse plane. Volume was estimated using the modified ellipsoidal formula V = ½(Length × Width^2^).

### 2.5. Glucoregulatory Assessments

A glucose tolerance test (GTT) was performed after 11 weeks of TRF. Mice were fasted for 6 h prior to the GTT, and fasting blood glucose was measured from a tail bleed using a glucometer (Easy Step Blood Glucose Monitoring System, Home Aide Diagnostics, Inc., Deerfield Beach, FL, USA). A bolus of 1 g/kg glucose was injected intraperitoneally, and blood glucose levels were measured at intervals up to 120 min. The glucose excursion curve was plotted over time, and the area under the concentration versus time curve from baseline (AUC glucose 0–120 min, mg/dL × minutes) was calculated. An insulin tolerance test (ITT) was performed at 12 weeks of TRF. Mice were fasted for 6 h prior to the ITT, and blood glucose was measured at 0 min, followed by injection of 0.65 U/kg insulin intraperitoneally. Blood glucose was measured at intervals up to 120 min. The glucose excursion curve was plotted over time and the area above the concentration versus time curve relative to baseline (AAC glucose 0–120 min, mg/dL × minutes) was calculated as insulin lowers glucose levels. A fasting insulin value was assessed in whole blood drawn prior to euthanasia using an ultra-sensitive Insulin ELISA kit (ALPCO). The homeostatic model of insulin resistance (HOMA-IR) was estimated as follows: (fasting plasma insulin concentration (mU/mL)) × (fasting blood glucose levels (mg/dL))/(405) [46].

### 2.6. Liver Histology

Liver tissue was harvested at ZT 6 (12 p.m.) following fasting from ZT 0 (6 a.m.) on the final day of the experiment. Hepatocellular steatosis was measured by Oil Red O staining on 7 μm cryosections of liver tissue [30]. Images were scanned by bright-field light microscopy (Eclipse TE300, Nikon, NY, USA) at ×20 magnification. The area of Oil Red O lipid staining was measured using Image J software (Version 1.54g) [47]. Hepatic steatosis was further assessed by hematoxylin and eosin (H&E) staining of 5 μm liver tissue sections of paraffin-embedded liver tissue. Images were scanned by a Nanozoomer 2.0HT Slide Scanner (Hamamatsu Photonics, Bridgewater, NJ, USA), and the number and area of lipid droplets (stained negative for H&E) were measured using Image J. Hepatic inflammation was assessed by counting infiltrating immune cells in the H&E-stained sections. Inflammation scoring was performed by a trained pathologist blinded to the sample identities. Liver fibrosis was assessed by Sirius Red staining and quantified by Image J. 

### 2.7. Quantification of Gene Expression by QPCR

RNA was isolated from frozen liver tissue using the RNA-STAT 60 reagent (AMSBIO LLC, Cambridge, MA, USA) and reversed transcribed into cDNA using the High-Capacity cDNA reverse transcription kit (Thermo Fisher Scientific, Waltham, MA, USA). Real-time PCR was performed on a Bio-Rad CFX384^TM^ real-time PCR detection system using iTaq^TM^ Universal SYBR^®^ Green Super mix (BioRad, Hercules, CA, USA). The primers are listed in Appendix A.

### 2.8. Tumor Histology

Mitotic counts were assessed on tumor sections stained with H&E. Tumor cell proliferation was further assessed by Ki67 staining of the tumor section. Slides of tumor sections were de-paraffinized with xylene and rehydrated followed by washing with distilled water. After incubation for 1 h with a blocking buffer (1% BSA in PBS), sections were incubated with a rabbit polyclonal anti-Ki67 antibody (ab15580, Abcam, Cambridge, UK, 1:2000 dilution, 2 h at RT). After washing three times with PBS-1% Tween20 (PBST), the sections were incubated with biotinylated secondary antibody for 30 min, followed by incubation with an avidin–biotin–peroxidase complex for 30 min at RT. Sections were washed with PBST and exposed to diaminobenzidine. Following washing, sections were counter stained with Meyer’s hematoxylin. Photographs of the section were captured using a Nanozoomer 2.0HT Slide Scanner (Hamamatsu Photonics, Bridgewater, NJ, USA). Ki-67 staining of liver and tumor sections from AL mice and liver sections from TRF were also performed as mentioned above.

### 2.9. Statistical Analysis

Data are presented as mean ± standard error of the mean. Data were analyzed by one-way or two-way ANOVA followed by multiple comparisons corrected using Tukey’s method. If these data were not normally distributed, they were analyzed by non-parametric tests, such as Mann–Whitney, Kruskal–Wallis, or Wilcoxon rank test. QPCR data were log2 transformed. All data were analyzed using Prism software (Version 10.1.1, Graphpad software, La Jolla, CA, USA), and the level of significance was set at *p* < 0.05.

## 3. Results

### 3.1. TRF Improved MASLD-Associated Metabolic Dysfunction without Caloric Restriction

We first evaluated the effect of TRF on metabolic dysfunction in obese mice on a MASH-inducing diet. The male mice were fed an ad libitum Western diet (WD) for 10 weeks before random assignment to continue in the ad libitum WD (AL) or time-restricted WD (TRF) groups using automated feeding cages (Figure 1A). The TRF group had access to food for 8 h (8 p.m.–4 a.m.) during the dark or active phase (Figure 1B). A control group of lean mice was fed ad libitum normal chow (NC). The AL group showed a rise in body weight over time, whereas the TRF group demonstrated an initial drop in body weight during the first week followed by a stabilization of body weight (Figure 1C) that was also reflected in a difference in body weight at euthanasia (Figure 1D). The mice on the NC diet remained lean throughout the study period. The obese AL mice had a doubling in visceral fat that was ameliorated in the TRF group, although they were still obese compared to the NC group (Figure 1E). Liver weight was also increased in the obese AL group but not the TRF group (Figure 1F). Food intake did not differ between the AL and TRF groups (Figure 1G), confirming that TRF is not associated with caloric restriction. Metabolic analysis using an intraperitoneal glucose tolerance test revealed the expected glucose intolerance in the obese AL group but improved glucose tolerance in the obese TRF group, whose glucose excursion curve was indistinguishable from the lean control mice (Figure 1H). The area under the glucose excursion curve confirmed this result (Figure 1I). Fasting blood glucose (FBG) levels were higher in the AL group, but the TRF group had levels equivalent to the lean control mice (Figure 1J). Obese AL mice were insulin intolerant by an insulin tolerance test (ITT) but TRF showed the same sensitivity as the lean NC mice (Figure 1K), which was confirmed by calculating the area above the glucose excursion curve (Figure 1L). Fasting plasma insulin was greatly elevated in the AL mice and reduced, but not normalized, by TRF (Figure 1M). A homeostatic model of insulin resistance (HOMA-IR) calculation showed insulin resistance in the AL group and improved insulin sensitivity in the TRF group (Figure 1N).

### 3.2. TRF Reduced Hepatic Steatosis in Obese MASH Mice

MASLD is characterized by hepatic steatosis. The livers of mice in the AL group were large and pale, suggesting steatosis, but the livers of TRF mice were normal size and had a normal appearance (Figure 2A). Inspection of H&E-stained liver sections indicated that the AL mice developed extensive macrovesicular and microvesicular steatosis (Figure 2A) that was reduced but not eliminated by TRF (Figure 2A). Pathological assessment of steatosis by a blinded pathologist indicated a decrease in steatosis score by TRF (Figure 2B). Quantification of lipid droplet number and size showed a decrease in both droplet number (Figure 2C) and size (Figure 2D) with a decrease in both microvesicular and macrovesicular lipid droplets (Figure 2E). Liver steatosis was further validated by Oil Red O staining, and histology and lipid droplet number and size supported the above finding (Figure 2F–H). At the gene expression level, the lipid biosynthetic genes *Fasn* and *Acc2*, the lipid transport gene *Cd36*, and the lipid storage genes *Cidea* and *Cidec* were significantly elevated in AL mice while TRF showed reduced expression, reflecting lower steatosis in the liver (Figure 2I–M). 

### 3.3. TRF Reduced Hepatic Inflammation and Fibrosis

The progression of MASLD to MASH is associated with hepatic inflammation and fibrosis. Livers of obese AL mice showed infiltration of inflammatory immune cells that was reduced in TRF mice by H&E staining (Figure 3A). Histological analysis of liver sections demonstrated that TRF reduced hepatic inflammation (Figure 3B). The expression levels of several macrophages and Kupffer cell markers (*Emr1*, *Clec4f*, and *Cd68*) were elevated in the AL group and reduced in the TRF group (Figure 3C–E); however, inflammatory cytokines (*Ccl2*, *Tnfa*, *Il6*, and *Il10*) were not significantly different (Figure 3F–I). Liver sections were stained with Sirius red to highlight fibrosis. Livers of AL mice showed extensive intralobular fibrosis that was absent in sections from lean mice (Figure 4A). Liver sections from TRF mice showed reduced fibrosis compared to the AL group (Figure 4A). Fibrosis was quantified by image analysis. Livers of AL mice had a significantly larger fibrotic area compared to NC and TRF mice (Figure 4B). The reduction in fibrosis by TRF was confirmed by QPCR. Livers of AL mice had an elevated expression of *Col1A1*, *Col3A1*, *Col4A1*, *Timp1*, *Fn1*, and *Acta2* (Figure 4C–H). The expression of *Col1A1*, *Col3A1*, *Col4A1*, and *Timp1* was decreased with TRF, but the expression of *Fn1* and *Acta2* was not significantly decreased (Figure 4C–H).

### 3.4. TRF Reduced Liver Cancer in a Spontaneous Genetic Model

We next examined the effect of TRF on liver tumor initiation and growth in a genetic model that develops tumors that are accelerated by MASH [43]. From 8 weeks of age, SRSF3-HKO mice were fed the MASH-inducing Western diet either ad libitum or with time-restricted access (Figure 5A). Ultrasound imaging of livers was performed after 16 weeks of TRF. Mice on the ad libitum WD showed intrahepatic tumors that were hypoechoic (Figure 5B) and were absent from the TRF mice (Figure 5C,D). Mice continued on the WD for a further 8 weeks before terminating this study. The SRSF3-HKO mice showed large pale livers (Figure 5E,F) and expanded adipose tissue (Figure 5E,G) consistent with our earlier observation in C57BL/6J mice (Figure 2). The livers of mice in the AL group had large visible tumors (Figure 5H–J), but no visible tumors were found in the livers of the TRF mice. Examination of liver sections from these mice revealed the presence of tumors in the AL mice and the complete absence of tumor foci in the TRF mice (Figure 5K). Cells in the tumors were characterized by macro and micro-vesicular steatosis with small irregular nuclei (Figure 5L). Hepatocytes in the non-tumor region of the AL livers showed normal nuclear morphology, but cellular organization was disrupted. Hepatocytes in the TRF livers showed the normal plate-like organization of hepatocytes. Immunohistochemical staining of liver and tumor sections from AL mice and liver sections from TRF mice showed increased Ki-67-positive cells in the tumor (Figure 5M,N). Staining also showed decreased proliferation of hepatocytes in TRF livers compared to AL livers.

### 3.5. TRF Reduces Growth of Hepa1-6 Tumors

To confirm the beneficial effect of TRF on liver cancer growth, we tested TRF in a subcutaneous mouse model of liver cancer. Eight-week-old male C57BL/6J mice were placed on a Western diet for 10 weeks to create MASH and then randomized to normal chow (NC), an ad libitum WD (AL), or a time-restricted WD (TRF) (Figure 6A). Four weeks later, Hepa1-6 mouse hepatocarcinoma cells were injected into the flanks of syngeneic C57Bl/6J mice. Tumor growth was measured over time using calipers. TRF abolished the tumor-promoting effect of the MASH-inducing diet such that tumor growth in the TRF group was indistinguishable from that in the NC group and notably lower compared to AL mice (Figure 6B). At the end of this study at 28 weeks, the tumor weight and volume were significantly higher in the AL group compared to the NC and TRF groups (Figure 6C,D). H&E-stained tumor sections showed increased mitotic figures in the tumors from mice in the AL group (Figure 6E,F) and increased Ki-67 staining (Figure 6G,H).

## 4. Discussion

The AASLD and AACE guidelines both recommend lifestyle modification for patients with MASLD with the addition of GLP-1 agonists only with evidence of MASH. Unfortunately, there are no approved drugs that directly treat MASLD or MASH, and weight loss is the first-line strategy for fatty liver disease. Recently, it has emerged that meal timing is important in maintaining healthy metabolism and reducing the risk of certain cancers. Eating at night is particularly detrimental as it is associated with a higher BMI, metabolic dysregulation, and increased risk of cancer [48,49,50,51]. In this regard, TRE (TRF in mice), which restricts eating to the normally active phase, improves metabolic health in animal models and improves cardiometabolic parameters in obese humans [22,23]. Daily fasting during Ramadan has a small beneficial effect on liver function, and a recent study reported a correction of liver enzymes, lipid profiles, and glycemic indices and also reported improvement in fibrosis markers (FIB4 and APRI) [39,40], but the effects are limited, as fasting occurs out of phase with normal circadian rhythms. Direct evidence for a positive effect of in-phase TRE on liver function comes from recent studies in patients with MASLD and MASH. A crossover trial of 32 individuals with MASLD showed that TRE decreased hepatic steatosis, weight, waist circumference, and BMI compared to standard care [52]. The TREATY-FLD study in 88 subjects with obesity and MASLD reported that TRE was as effective as caloric restriction in lowering intrahepatic triglyceride and liver stiffness [37]. Other forms of intermittent fasting, such as alternate day fasting (ADF), have also been shown to reduce hepatic steatosis and liver stiffness [53,54,55]; however, these ADF studies include calorie restriction during the fasting days, and calorie restriction alone can improve liver function. The TRE findings in the absence of calorie restriction underscore the importance of meal timing that is synchronized to circadian rhythms to maximize improvement in overall metabolism. Studies of TRE also report high adherence and long-term weight loss maintenance, as TRE does not involve calorie counting or complex meal planning [22,23,56]. As the TRE fasting cycle occurs during the night allowing an 8–10 h eating window during the day with no calorie restriction, it may require less cognitive effort and facilitate dietary satisfaction. Furthermore, TRE may decrease conflict with the homeostatic drive to eat and prevent dietary intervals with prolonged negative energy balance [16]. A small study of TRE for 12 weeks involving 19 individuals with metabolic syndrome revealed enhanced metabolic health and, importantly, reported good long-term compliance with TRE intervention months after the end of the trial, and many participants were still practicing TRE and sustained their weight loss one year after the end of this study [57,58]. Furthermore, in a large, randomized controlled trial of TRE in 116 overweight and/or obese men and women, great adherence to the TRE regimen (8 h feeding window) was reported [59]. The beneficial effect is borne out by epidemiological studies. In a cross-sectional study of 3813 participants, a daily eating window of <8 h was associated with reduced MASLD [38], but there was no association with physical activity or diet quality.

Despite the positive metabolic results, there remains a paucity of studies of TRE in cancer patients. Weight loss through caloric restriction and exercise has been reported to be beneficial in preventing cancer. Sustained weight loss over 10 years reduces the risk of breast cancer, whereas stable weight or short-term weight loss over one 5-year interval does not reduce risk [60]. This observation underscores the necessity for an intervention that is sustainable over a prolonged period. Indeed, although caloric restriction has shown promising results in various cancers, including HCC, long-term compliance is challenging as most people regain the lost weight. A TRE study in 22 overweight women with early-stage breast cancer reported decreases in fat mass and improved nutritional status, demonstrating the feasibility of this intervention in cancer patients [61]. Mechanistically, disruption of normal circadian rhythms can induce cancer, including HCC, in various preclinical models [62], and clock components control cell cycle genes and proliferation [38], leading to the suggestion that improving circadian rhythms by time-restricted feeding could inhibit obesity-driven cancer. Indeed, this has been demonstrated in a mouse model of obese postmenopausal breast cancer where TRF reduced cancer initiation and growth, and metastasis to the lung [23]. Similarly, meal timing from 8 a.m. to 12 p.m. resulted in reduced tumor growth in a model of pancreatic adenocarcinoma [63], and restricting food intake to 6 h inhibits lung cancer progression [64,65]. Despite these findings, there have been no interventional studies in animals or humans to study the effect of TRF on MASH-driven liver cancer. 

The molecular mechanisms underlying the reported beneficial effects need to be explored further. One possible mechanism could be TRF acting on peripheral clocks, which are exquisitely sensitive to the fasting–feeding cycle. At the molecular level, fasting elevates the AMP/ATP ratio, activating AMPK, which phosphorylates the serine71 of CRY1, reducing its stability [66]. AMPK also regulates Casein kinase I epsilon, another kinase crucial for PER phosphorylation and stability [67]. Similarly, fasting activates NAMPT, increasing NAD+ levels and SIRT1 activity. SIRT1 interacts with CLOCK:BMAL1 and suppresses *Per2* transcription [68]. Conversely, feeding activates mTOR, inducing CRY1 post-transcriptionally [69]. Thus, fasting boosts the positive limb (CLOCK and BMAL1) while feeding enhances the negative limb (CRY and PER) of the circadian clock. Importantly TRF is able to modulate the expression and activity of these signaling pathways and clock components [70]. 

Other clock-independent mechanisms are also possible. Transcriptome and metabolomics studies in the liver of mice that have an intact or dysfunctional circadian clock have revealed that integrated stress response (ISR) pathways may contribute to TRF benefits [71]. TRE also affects gut function. A survey of gut microbiome composition across 24 h in mice revealed that TRF decreases the relative amounts of presumed obesogenic microflora and increases the relative amounts of presumed obesity-protective microflora [72]. Furthermore, the stool of TRF mice was rich both in primary and secondary bile acids [72], and the elevated luminal concentration of bile acids might also affect bile acid signaling in the liver. TRE also reduces oxidative stress and inflammation. A study investigated the impact of a 6-week TRE intervention (14:10; fasting/feeding) in resistance-trained firefighters [73]. Time-restricted eating resulted in significant reductions in advanced oxidation protein products, suggesting a beneficial effect on redox homeostasis. Further studies will be needed to dissect the relative importance of these molecular mechanisms.

## 5. Limitations of this Study

This study only tested male C57BL6/J mice, as female mice are relatively protected from obesity and insulin resistance, and C57BL/6 mice are known to be sensitive to obesity-induced metabolic dysregulation. Furthermore, this study only tested two liver cancer models, and mouse models of HCC often lack features of human HCC, so extrapolation to humans might be premature. This is compounded by the multiple etiologies of liver cancer development in humans. It is also possible that spontaneous tumors might eventually form in the SRSF3 KO mice on TRF even though the acceleration due to MASH is prevented. Further studies will be needed to show if long-term cancer protection is afforded by TRF or whether TRF itself has detrimental effects over extended periods.

## 6. Conclusions

In the present study, we report that TRF improved liver and metabolic health in obese male mice on a MASH-inducing Western diet without altering calorie intake. We confirmed that TRF normalized hyperinsulinemia and improved metabolic dysregulation. TRF decreased lipid accumulation, inflammation, and fibrosis, which are hallmarks of MASH [74]. Furthermore, we showed that TRF resulted in a striking inhibition of tumor initiation and tumor progression compared to mice with ad libitum access to the Western diet. These findings support the possibility that simple dietary manipulation through altered meal timing might be a practical way to reduce cancer risk for cancers that are associated with obesity. On the basis of our preliminary findings, clinical trials to determine the relevance of TRE to human MASH and HCC are warranted. Thus, TRE could be a potential non-pharmacological intervention to prevent or inhibit MASH and the progression of MASH-driven liver cancer.

## Figures and Tables

**Figure 1 cancers-16-01513-f001:**
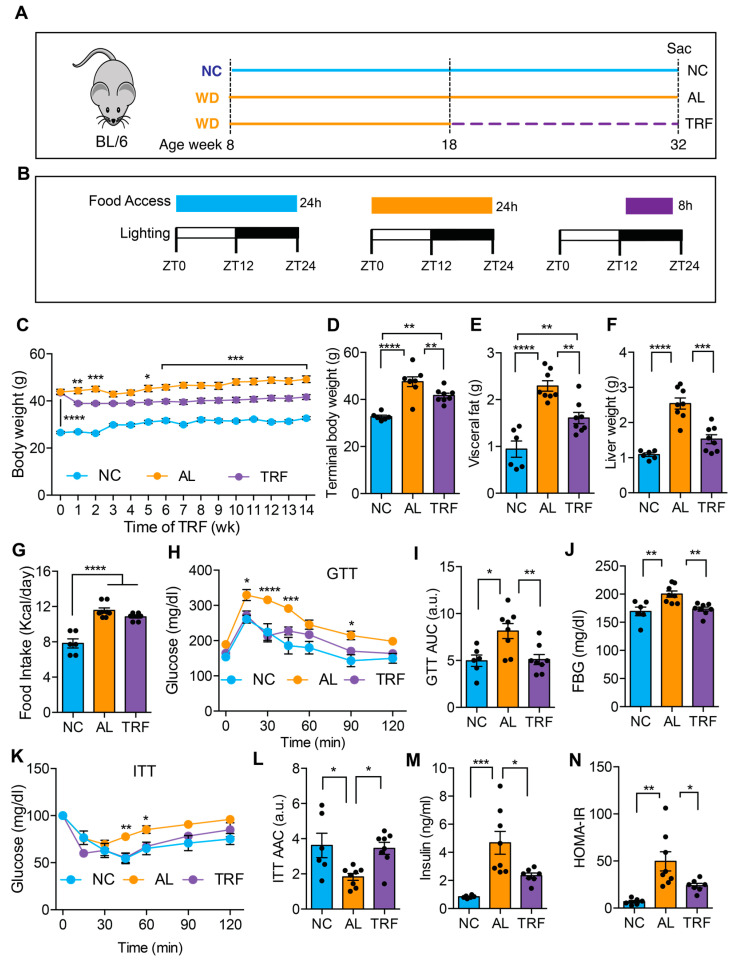
TRF reduced body weight and improved metabolic parameters in a Western diet-fed mouse MASH model. (**A**) Schematic of the TRF of a Western diet (WD)-fed mouse model and feeding groups of normal chow (NC), Western diet ad libitum (AL), and WD time-restricted feeding (TRF) used in this study. NC mice are colored blue, AL mice orange, and TRF mice purple. The dotted purple line indicates the period of TRF. (**B**) The diet, the food access time, and the lighting schedule are shown. ZT0 is light on at 6 a.m. and ZT12 is light off at 6 p.m. (**C**) Weekly body weights of mice on ad libitum NC, ad libitum WD, and time-restricted WD after initiation of the TRF protocol (n = 6 for NC, n = 8 for AL, n = 8 for TRF). NC mice are colored blue, AL mice orange, and TRF mice purple in all figure panels. (**D**–**F**) Terminal body, visceral fat weight, and liver weight. (**G**) Daily mean food intake in mice (kcal/day). (**H**) Intraperitoneal glucose tolerance test (GTT) after 11 weeks of TRF. (**I**) Area under the curve (AUC) for GTT assay. (**J**) Fasting blood glucose (FBG) levels at the end of this study. (**K**) Intraperitoneal insulin tolerance test (ITT) after 12 weeks of TRF. (**L**) Area above the curve for ITT assay. (**M**) Fasting insulin levels at the end of this study. (**N**) Homeostatic model assessment of insulin resistance (HOMA-IR). All data are presented as mean ± SEM with n representing the number of mice per group. Asterisks show a statistical significance of TRF vs. AL or as shown. * *p* < 0.05, ** *p* < 0.01, *** *p* < 0.001, **** *p* < 0.0001.

**Figure 2 cancers-16-01513-f002:**
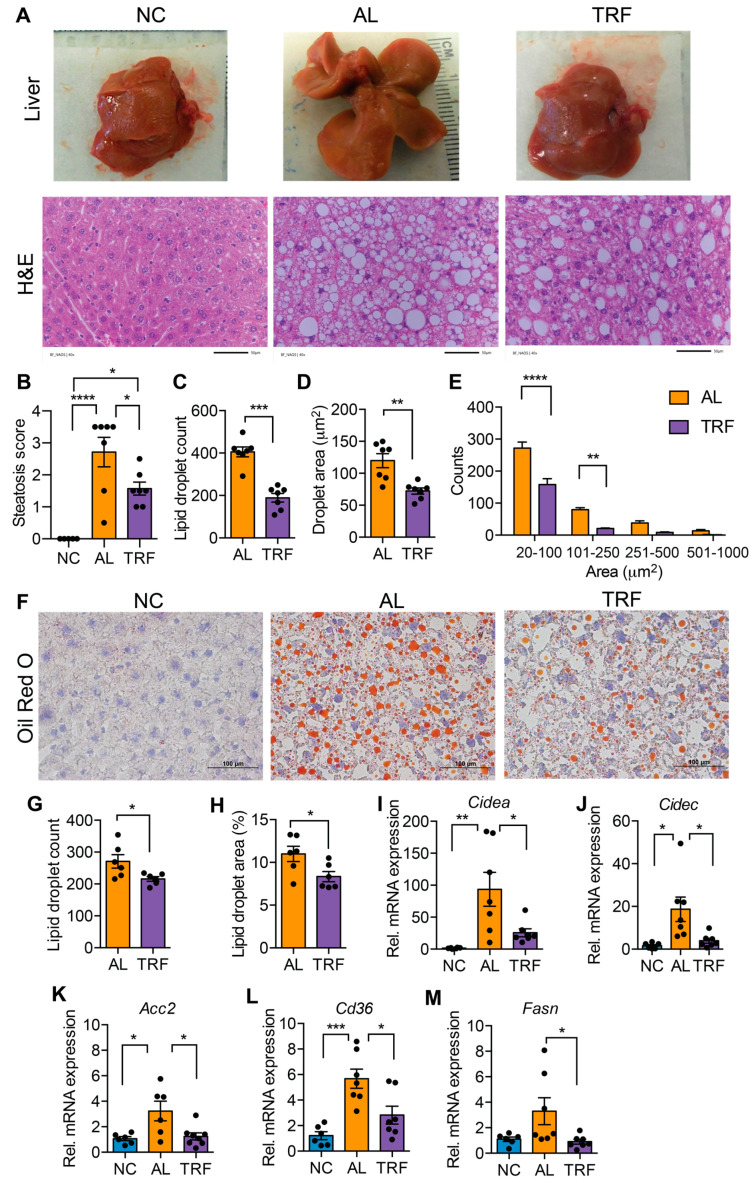
TRF improves liver steatosis and function. (**A**) Gross liver morphology at the end of this study and H&E staining of liver section of mice on NC, AL, and TRF showing hepatic steatosis. Scale bar represents 50 μm. (**B**) Steatosis scoring of the H&E-stained liver section. (**C**–**E**) Lipid droplet counts and area measurements on liver sections. (**F**) Oil Red O staining of liver sections of mice on NC, AL, and TRF. Scale bar represents 100 μm. (**G**,**H**) Lipid droplet counts and area measurements on liver sections. (**I**–**M**) Analysis of lipid synthesis, transport, and storage genes in the liver tissue of mice of NC, AL, and TRF. Color scheme is the same as Figure 1. All data are presented as mean ± SEM with n representing the number of mice per group. Asterisks show a statistical significance of TRF vs. AL or as shown. * *p* < 0.05, ** *p* < 0.01, *** *p* < 0.001, **** *p* < 0.0001.

**Figure 3 cancers-16-01513-f003:**
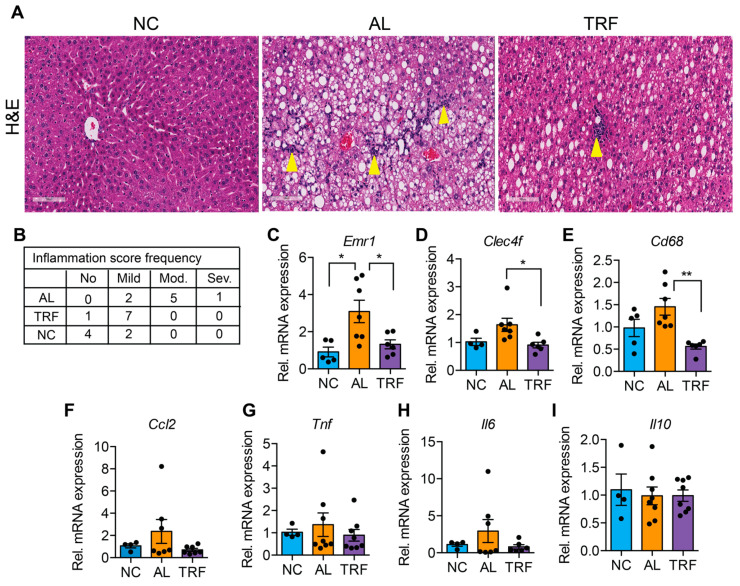
TRF improves liver inflammation. (**A**) H&E staining of the liver section showing immune cell infiltration (yellow arrowhead). Scale bar equals 100 µm. (**B**) Inflammation scoring by a blinded pathologist. (**C**–**I**) Analysis of inflammatory genes in the liver of mice on NC, AL, and TRF. Color scheme is the same as Figure 1. All data are presented as mean ± SEM with n representing the number of mice per group. Asterisks show a statistical significance of TRF vs. AL or as shown. * *p* < 0.05, ** *p* < 0.01.

**Figure 4 cancers-16-01513-f004:**
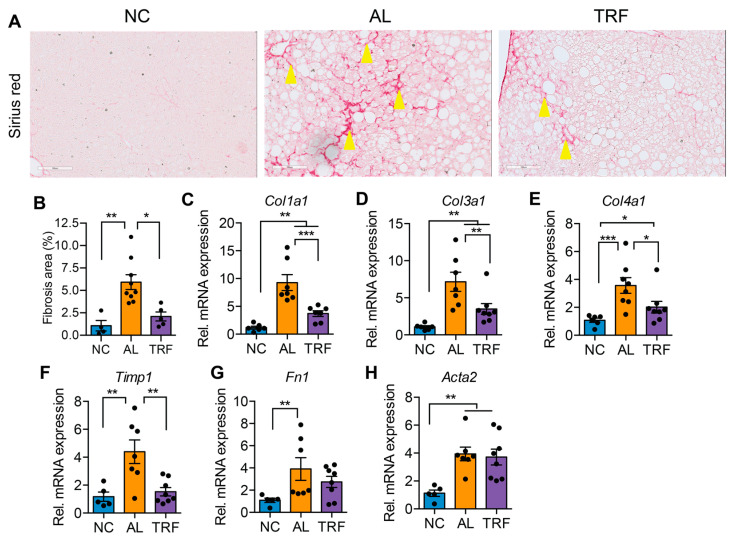
TRF improved liver fibrosis. (**A**) Sirius red staining of liver sections showing fibrosis (stained red, yellow arrowheads). Scale bar equals 100 µm. (**B**) Fibrosis area quantification by Image J software. (**C**–**H**) Analysis of fibrosis-associated gene expression in livers of mice on NC, AL, and TRF. Color scheme is the same as Figure 1. All data are presented as mean ± SEM. Asterisks show a statistical significance of TRF vs. AL or as shown. * *p* < 0.05, ** *p* < 0.01, *** *p* < 0.001.

**Figure 5 cancers-16-01513-f005:**
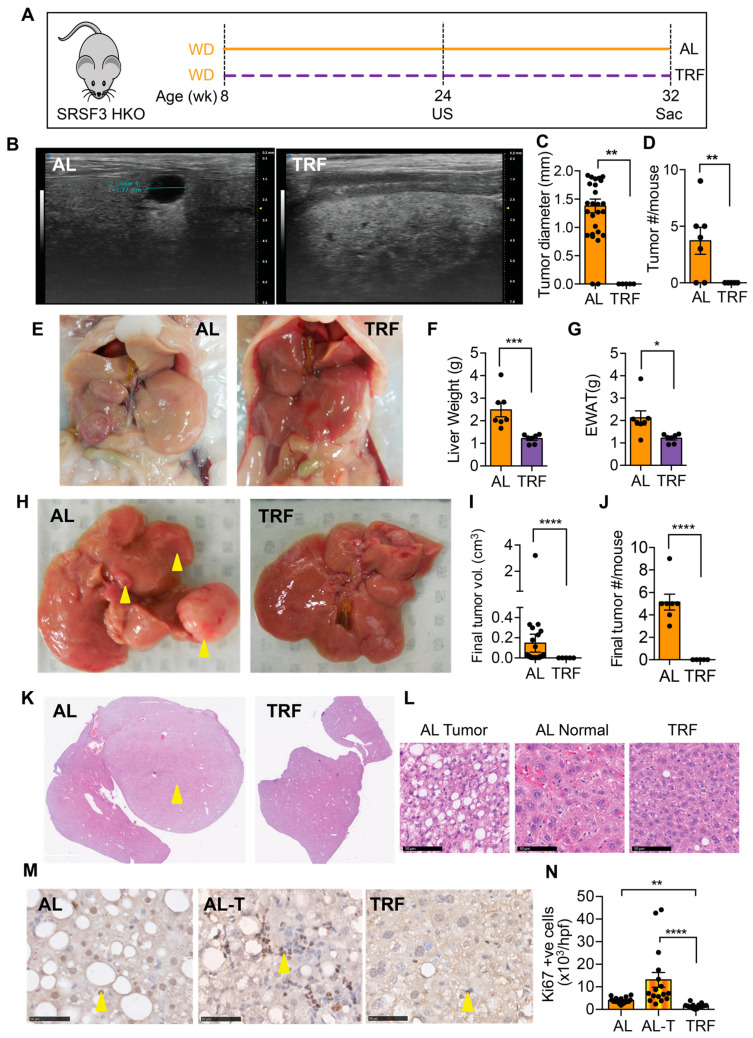
TRF attenuated tumor initiation and growth in a SRSF3 hepatocyte knockout (SRSF3-HKO) mouse model of liver cancer. (**A**) Scheme of TRF in Western diet-fed SRSF3-HKO mice. (**B**) Tumor analysis by ultrasound (US) imaging of the liver at 24 weeks. (**C**,**D**) Tumor size and number from US imaging. (**E**) Representative gross liver and adipose tissue morphology at the end of this study at 32 weeks. (**F**,**G**) Terminal liver and visceral fat weight. (**H**) Representative livers showing tumor at the time of sacrifice (yellow arrowheads). (**I**,**J**) Tumor volume and numbers at sacrifice. (**K**) H&E staining of liver sections showing tumors (yellow arrowhead) in the AL group. (**L**) Cellular morphology in tumors and non-tumor tissue. Scale bar equals 50 µm. (**M**) Ki-67 staining in sections from the AL liver, AL tumor (AL-T), or TRF liver. Arrowheads indicate positive cells. Scale bar equals 50 µm. (**N**) Quantification of Ki-67 staining. Asterisks show a statistical significance of TRF vs. AL or as shown. * *p* < 0.05, ** *p* < 0.01, *** *p* < 0.001, **** *p* < 0.0001.

**Figure 6 cancers-16-01513-f006:**
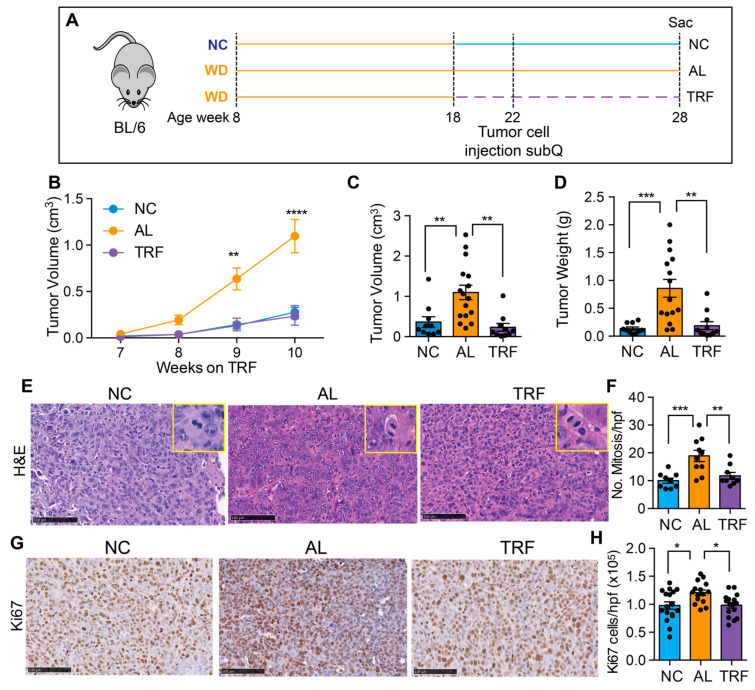
TRF reduced tumor growth in a subcutaneous mouse model of liver cancer. (**A**) Scheme of TRF in a MASH diet-induced subcutaneous mouse model of liver cancer. (**B**) Hepa1-6 tumor volume over time in mice on NC, AL, and TRF. (**C**) Individual tumor volume at the end of this study. (**D**) Individual tumor weight at the end of this study. (**E**) Representative image of H&E staining of Hepa1-6 tumor tissue. Scale bar represents 100 μm. Insets show higher power images of cells with mitoses. (**F**) Quantification of mitoses. Data are presented as mean ± per high-powered field (hpf). (**G**) Representative image of Ki67 staining of Hepa1-6 tumor tissue. Scale bar represents 100 μm. (**H**) Quantification of Ki67 positive cells. Data are presented as mean ± per high-powered field (hpf). Asterisks show a statistical significance of TRF vs. AL or as shown. * *p* < 0.05, ** *p* < 0.01, *** *p* < 0.001, **** *p* < 0.0001.

## Data Availability

All the data supporting the findings of this study are available within the article and its Appendix A and from the corresponding author upon reasonable request.

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
