# Peer review of "Time-Restricted Feeding Attenuates Metabolic Dysfunction-Associated Steatohepatitis and Hepatocellular Carcinoma in Obese Male Mice"

_cancers, 2024, doi:10.3390/cancers16081513_

Round 1

Reviewer 1 Report

Comments and Suggestions for Authors

I gladly reviewed the manuscript ID: cancers-2928542, entitled as Time-Restricted Feeding Attenuates Metabolic Dysfunction-Associated Steatohepatitis and Hepatocellular Carcinoma in Obese Mice " submitted by Das, M. et al. The idea is practical, the results are informative and the layout is clearly presented. Nonetheless, some concerns listed as follows need more clarification:

1.     The body weight, visceral fat and liver weight of mice on western diet (WD) is indeed gained, and there is more fat droplets of liver in histologic presentation as compared to those on normal chow diet (NC). However, can we call the increases in these areas as “obesity” in mice? In my own perspective, “overweight” could be better fit.

2.     Is MASH-induced change equivalent to obesity-related change? If uncertain, please revise the title.

3.     This is a male mice model, so rephrase the title.

4.     How long do the mice spontaneously develop HCC following the initiation of WD in your study design with no cancer cell line xenograft?

5.     Do the mice treated with TRF grow tumor again or change inflammation pattern if discontinue TRF and resume WC or NC in both xenograft models?

6.     VevoMD ultrasound could measure the liver mass volume but the real number of liver nodules? The severity of liver lesion should be assessed by size and number simultaneously.

7.     Why pick up these two HCC cell lines (SRFS-HKO and Hepa1-6) in this study?

8.     Is there any limitation in this study?

Author Response

Reviewer1:  I gladly reviewed the manuscript ID: cancers-2928542, entitled as” Time-Restricted Feeding Attenuates Metabolic Dysfunction-Associated Steatohepatitis and Hepatocellular Carcinoma in Obese Mice " submitted by Das, M. et al. The idea is practical, the results are informative and the layout is clearly presented.

Comment1: The body weight, visceral fat and liver weight of mice on western diet (WD) is indeed gained, and there are more fat droplets of liver in histologic presentation as compared to those on normal chow diet (NC). However, can we call the increases in these areas as “obesity” in mice? In my own perspective, “overweight” could be better fit.

Response:  Based on the previous studies, the increase in body weight, visceral fat, liver health and lipid accumulation in liver are considered as the criteria to define obesity (see for reference: Diet-induced obesity in animal models: points to consider and influence on metabolic markers, Diabetol Metab Syndr. 2021). In humans, obesity is generally defined by BMI but that does not translate to mice. An easier comparison is using body fat percentage. An obese man has >25% body fat (30% for women). A lean male mouse is 25-30 g.  We generally consider mice over 40 g as being obese, these mice would have a body fat percentage of 25% assuming a lean mass of 30g.  This is consistent with other labs. So, we feel that using the term obese is legitimate in our study.

Comment2. Is MASH-induced change equivalent to obesity-related change? If uncertain, please revise the title.

Response:  Thank you for your suggestion. We have previously published the beneficial effect of TRF in a high fat feeding model of obesity with simple steatosis. This paper shows a similar beneficial effect on a MASH model. The weight gain is not as great with the MASH model compared to the simple HFD model, but the mice are still considered obese. MASH is obviously more relevant to HCC development, so we have changed the title as suggested.

Comment3. This is a male mice model, so rephrase the title.

Response: As per the reviewer suggestion we rephrase the title of the revised MS

Comment4. How long do the mice spontaneously develop HCC following the initiation of WD in your study design with no cancer cell line xenograft?

Response: In our previous study we reported that lean SRSF3-KO mice on normal chow diet develop tumors at 12-14 months of age (see for reference: Deletion of Splicing Factor SRSF3 in Hepatocytes Predisposes to Hepatocellular Carcinoma in Mice, Hepatology. 2015). Tumor formation is accelerated in the MASH model. We put the SRSF3-KO mice on WD at age of 8 weeks and observed the initiation of tumor nodules after 16 weeks of WD (at age 24 weeks) so by tumors after detected by 6 months instead of 12-14 months for normal chow. 

Comment5. Do the mice treated with TRF grow tumor again or change inflammation pattern if discontinue TRF and resume WC or NC in both xenograft models?

Response: We have not conducted a study to observe tumor growth in mice subjected to such a diet switch but that is a good idea for future studies. However we have performed a yo-yo diet switch study with mice on a HFD. In this case, body weight drops to the level of lean normal chow mice within 8 weeks of cessation of the HFD. If the mice are then placed back on HFD they rapidly regain the lost weight to match mice left continually on HFD. So that would suggest that the detrimental metabolic effects would come back quickly but we don't know how that would translated to tumor growth.

Comment 6. VevoMD ultrasound could measure the liver mass volume but the real number of liver nodules? The severity of liver lesion should be assessed by size and number simultaneously.

Response: The tumor lesion has been quantified for size and numbers from US images and presented in figure 5C, and 5D.

Comment 7. Why pick up these two HCC cell lines (SRFS-HKO and Hepa1-6) in this study?

Response:

Hepa1-6 is a murine hepatoma cell line that was originally derived from a spontaneous hepatoma in a C57BL/6 mouse and has been extensively characterized for its growth properties, gene expression profile, and response to various anti-cancer agents. They are highly tumorigenic in immunocompetent C57BL/6 mice, and can form tumors when injected subcutaneously or orthotopically into the liver. Compatibility with C57BL/6 mice is important as this is the most commonly used strain for obesity studies.  Xenograft studies in immunocompromised mice would not be subject to the same immunosurveillance as immunocompetent mice.

The SRSF3 KO model is attractive as we have published that this genetic deletion disrupts liver metabolism, increases ER stress, causes a chronic hepatocyte damage/proliferation phenotype, eventually leading to HCC.  The resulting tumors have mutational signatures similar to human HCC and rely on IGF2 autocrine signaling for proliferation and tumor formation. We also showed that SRSF3 protein is reduced in human liver disease (MAFLD, MASH, cirrhosis, and HCC) and alteration of SRSF3 targets is seen in HCC and is predictive of survival. So we believe this model reflects many aspects of human liver disease and HCC. .

Comment 8. Is there any limitation in this study?

Response: We have added a limitations paragraph to the manuscript as suggested by multiple reviewers. In addition to investigating the impact of time-restricted feeding (TRF) on MASH and HCC, we aimed to delve into its mechanistic link. However, due to the limited availability of automated TRF cages and the substantial time required for each experiment (typically 20-25 weeks), we have deferred the exploration of its mechanism to our future endeavors. Nevertheless, our study offers valuable insights into the involvement of TRF in obesity-driven liver cancer, establishing a foundation for future research in this domain."

Reviewer 2 Report

Comments and Suggestions for Authors

In this work, Das and collaborators evaluate the impact of time-restricted feeding (TRF) on the development of MASH and hepatocellular carcinoma (HCC) in different mouse models of MASLD and HCC. This is a little-explored area of high relevance in the field of chronic liver disease and carcinogenesis. The authors observe that the progression of MASLD is attenuated upon the implementation of TRF in animals fed a Western diet (WD) without alterations in food intake. TRF also blunts the development of HCC in a genetic mouse model (SRSF3-KO mice) fed a WD, as well as the growth of subcutaneously implanted murine HCC cells in immune-competent mice. This is a properly designed study with clear and relevant data in the field. The results are robust and very well presented. I only have a few suggestions for the authors to consider.

1. As the authors mention, further mechanistic studies that may explain the phenotypes observed are warranted. While these studies are going on, could the authors further discuss potential mechanisms underlying the relevant differences found here? Could TRF affect the levels of oxidative stress in the liver, growth factors expression, gut permeability, etc?

2. Bile acids (BA) metabolism follows a circadian rhythm, and their hepatic accumulation in MASLD can trigger liver injury, eventually contributing to hepatocarcinogenesis. It may be interesting to evaluate intrahepatic BA levels in these mouse models.

3. It would be interesting to evaluate the degree of hepatocellular proliferation (Ki-67 staining) in the WD-MASLD and SRSF3-KO AL/TRF models.

4. In the abstract, the authors mention that TRF abrogated metabolic dysregulation induced by WD without any weight loss. If I understood it correctly, TRF mice showed reduced body weight compared to AL-fed mice (Fig. 1D). Please clarify.

5. It would be helpful to better explain why the SRSF3-KO mouse model was selected for this study. Could the findings obtained in this model be expected in other (genetic) models of liver injury and carcinogenesis?

Author Response

Reviewer 2:  In this work, Das and collaborators evaluate the impact of time-restricted feeding (TRF) on the development of MASH and hepatocellular carcinoma (HCC) in different mouse models of MASLD and HCC. This is a little-explored area of high relevance in the field of chronic liver disease and carcinogenesis. The authors observe that the progression of MASLD is attenuated upon the implementation of TRF in animals fed a Western diet (WD) without alterations in food intake. TRF also blunts the development of HCC in a genetic mouse model (SRSF3-KO mice) fed a WD, as well as the growth of subcutaneously implanted murine HCC cells in immune-competent mice. This is a properly designed study with clear and relevant data in the field. The results are robust and very well presented. I only have a few suggestions for the authors to consider.

Comment 1. As the authors mention, further mechanistic studies that may explain the phenotypes observed are warranted. While these studies are going on, could the authors further discuss potential mechanisms underlying the relevant differences found here? Could TRF affect the levels of oxidative stress in the liver, growth factors expression, gut permeability, etc?

Response: Thank you for your valuable suggestion. As suggested, we have added a discussion of potential mechanisms underlying the beneficial effect of TRF in conclusion section (page 16-18).

Comment 2. Bile acids (BA) metabolism follows a circadian rhythm, and their hepatic accumulation in MASLD can trigger liver injury, eventually contributing to hepatocarcinogenesis. It may be interesting to evaluate intrahepatic BA levels in these mouse models.

Response: We think that is a great suggestion specially with the interest in secondary bile acids from the gut. Bile acids, lipid or cholesterol metabolites, ketone bodies are all potential mediators of a beneficial effect, and future studies are planned to evaluate the involvement of these in the beneficial effect of TRF.

Comment 3. It would be interesting to evaluate the degree of hepatocellular proliferation (Ki-67 staining) in the WD-MASLD and SRSF3-KO AL/TRF models.

Response: Per your suggestion we performed the Ki-67 staining SRSF3-KO AL/TRF models and the results have been incorporated in Figure 5 (M, N), page 13, Para 2.

Comment 4. In the abstract, the authors mention that TRF abrogated metabolic dysregulation induced by WD without any weight loss. If I understood it correctly, TRF mice showed reduced body weight compared to AL-fed mice (Fig. 1D). Please clarify.

Response: The reviewer is correct. When mice are placed on TRF, there is an initial slight loss in body weight as the mice become accustomed to the new eating schedule, then a stabilization of body weight (Figure 1C).  The body weight does not drop after the initial dip and the mice ae still in the obese range ~40g. The mice in the AL group on the other hand continue to overeat and put on more weight.  So, the weights start to diverge after 4 weeks. Importantly, the body weight in the TRF group does not change from week 1 although the metabolic assessment continues to improve over time.

Comment 5. It would be helpful to better explain why the SRSF3-KO mouse model was selected for this study. Could the findings obtained in this model be expected in other (genetic) models of liver injury and carcinogenesis?

Response: Please see response to reviewer 1. As per your suggestion we edited the text to explain on the choice of the SRSF3-KO mouse model for this study (Page 6). We suspect that TRF would be beneficial in other model as well although we have not performed those studies. TRF will likely only be beneficial in models that are accelerated by obesity.  Along those lines, TRF has no effect in an model of liver cancer using RIL-175 cells, but these cells are highly transformed and growth of tumors is not affected by obesity so they would not be expected to respond to TRF.

Reviewer 3 Report

Comments and Suggestions for Authors

I’ve read with attention the paper of DAS et al. that is potentially of interest. The background and aim of the study have been clearly defined. The methodology applied is overall correct, the results are reliable and adequately discussed. The conclusions are consistent with the evidence and arguments presented and they address the main question posed. The references are also appropriate as well as tables and figures. I have no ethical concerns regarding experiments, nor on plagiarism or publication ethics. I’ve only some minor comments:

- The abstract is not well-balanced, with a very long introduction, and very short methods/results sections. This can be improved

- The background ends with a conclusion in spite of the aim of the study. This should be corrected

- The number of mice used in the text should be immediately included at the beginning of the method section

- The statistical analysis has been carried out as all the investigated parameters have a normal distribution (that is unlikely). This has to be tested and then the statistics improved

- The limitations of the proposed experimental approach should be listed and shortly commented

Comments on the Quality of English Language

No concerns. Only some small typos.

Author Response

Reviewer 3: I’ve read with attention the paper of DAS et al. that is potentially of interest. The background and aim of the study have been clearly defined. The methodology applied is overall correct, the results are reliable and adequately discussed. The conclusions are consistent with the evidence and arguments presented and they address the main question posed. The references are also appropriate as well as tables and figures. I have no ethical concerns regarding experiments, nor on plagiarism or publication ethics. I’ve only some minor comments: -

Comment 1. The abstract is not well-balanced, with a very long introduction, and very short methods/results sections. This can be improved.

We have edited the abstract as suggested.

Comment 2. The background ends with a conclusion in spite of the aim of the study. This should be corrected –

Response: Thank you for your suggestion and as suggested we edited the background (page 5, para 2)

Comment 3. The number of mice used in the text should be immediately included at the beginning of the method section

Response: Thank you for your suggestion and we edited the revised MS for the use of number of mice (page 5, para 3, page 7, para 2,).

Comment 4.  The statistical analysis has been carried out as all the investigated parameters have a normal distribution (that is unlikely). This has to be tested and then the statistics improved –

Response: We do check for normality using PRISM although we did not include the information in the text. If data does not seem to be normally distributed we use non-parametric test such as Wilcoxon rank. We edited the revised MS accordingly.

Comment 5. The limitations of the proposed experimental approach should be listed and shortly commented

We have added a. limitations section to the manuscript.

Reviewer 4 Report

Comments and Suggestions for Authors

The manuscript presented by Manasi Das and co-workers. title: “Time-Restricted Feeding Attenuates Metabolic Dysfunction Associated Steatohepatitis and Hepatocellular Carcinoma in Obese Mice” is well written, clear, and easy to read. The topic is interesting, and therefore, it adds clustered information to the subject area of the type concerning diet (in this case, westernized one, rich in sugar and fat) and the influence of it on the burden condition of metabolic syndrome. Moreover, this is the first study, to my knowledge, that links TRF and liver cancer. So, no other published articles are present in the literature in this sense.

Said that it is helpful to add a section taking into account the “limitation” to translating this concept easily to humans since there is concern about silent HBV infection and liver cancer development. On this focus, HVB liver infection is not easy to assess, and microRNAs liver-specific could help. 

Please consider the following reference:

1-    https://doi.org/10.3390/ijms25020975

2-    10.3389/fcimb.2022.790964

3-    https://doi.org/10.3390/cimb45010006

Author Response

Reviewer 4: The manuscript presented by Manasi Das and co-workers. title: “Time-Restricted Feeding Attenuates Metabolic Dysfunction Associated Steatohepatitis and Hepatocellular Carcinoma in Obese Mice” is well written, clear, and easy to read. The topic is interesting, and therefore, it adds clustered information to the subject area of the type concerning diet (in this case, westernized one, rich in sugar and fat) and the influence of it on the burden condition of metabolic syndrome. Moreover, this is the first study, to my knowledge, that links TRF and liver cancer. So, no other published articles are present in the literature in this sense.

Comment 1: Said that it is helpful to add a section taking into account the “limitation” to translating this concept easily to humans since there is concern about silent HBV infection and liver cancer development. On this focus, HVB liver infection is not easy to assess, and microRNAs liver specific could help. Please consider the following reference: 1- https://doi.org/10.3390/ijms25020975 2- 10.3389/fcimb.2022.790964 3- https://doi.org/10.3390/cimb45010006.

Response: Thank you for your valuable suggestion, and as per your suggestion we added a paragraph discussing the challenges in translating TRE to humans. (Page 18, Para 2)

Round 2

Reviewer 3 Report

Comments and Suggestions for Authors

The authors have deeply revised their paper according to the useful comments of the reviewers. I have no futher comments on it.